nanotechnology

graphene nanosheets, mechanochemical, exfoliation

**Author for correspondence:**
Nicole N. Hashemi
e-mail: nastaran@iastate.edu

This article has been edited by the Royal Society of Chemistry, including the commissioning, peer review process and editorial aspects up to the point of acceptance.

# Protein-assisted scalable mechanochemical exfoliation of few-layer biocompatible graphene nanosheets

Deepak-George Thomas[1], Steven De-Alwis[1],
Shalabh Gupta[2], Vitalij K. Pecharsky[2,3],
Deyny Mendivelso-Perez[2,4], Reza Montazami[1],
Emily A. Smith[2,4] and Nicole N. Hashemi[1,5]

[1]Department of Mechanical Engineering, Iowa State University, Ames, IA 50011-2030, USA
[2]The Ames Laboratory, US Department of Energy, Ames, IA 50011-3020, USA
[3]Department of Material Science and Engineering, Iowa State University, Ames, IA, 50011-1096, USA
[4]Department of Chemistry, Iowa State University, Ames, IA, 50011-1021, USA
[5]Department of Biomedical Sciences, Iowa State University, Ames, IA 50011, USA

RM, 0000-0002-8827-0026; NNH, 0000-0001-8921-7588

A facile method to produce few-layer graphene (FLG) nanosheets is developed using protein-assisted mechanical exfoliation. The predominant shear forces that are generated in a planetary ball mill facilitate the exfoliation of graphene layers from graphite flakes. The process employs a commonly known protein, bovine serum albumin (BSA), which not only acts as an effective exfoliation agent but also provides stability by preventing restacking of the graphene layers. The latter is demonstrated by the excellent long-term dispersibility of exfoliated graphene in an aqueous BSA solution, which exemplifies a common biological medium. The development of such potentially scalable and toxin-free methods is critical for producing cost-effective biocompatible graphene, enabling numerous possible biomedical and biological applications. A methodical study was performed to identify the effect of time and varying concentrations of BSA towards graphene exfoliation. The fabricated product has been characterized using Raman spectroscopy, powder X-ray diffraction, transmission electron microscopy and scanning electron microscopy. The BSA-FLG dispersion was then placed in media containing Astrocyte cells to check for cytotoxicity. It was found that lower concentrations of BSA-FLG dispersion had only minute cytotoxic effects on the Astrocyte cells.

# 1. Introduction

Pristine graphene is a two-dimensional material consisting of carbon atoms, hexagonally arranged, exhibiting $sp^2$ hybridization and forming a sheet possessing the thickness of a single atom [1,2]. Graphene possesses remarkable electrical, mechanical and thermal properties, attributed to its π-conjugation [1,3,4]. The charge carrier mobility of freely suspended graphene exceeds 2000 $cm^2 V^{-1} s^{-1}$ [5], accredited to graphene's outstanding electrical properties [6]. Despite being the thinnest material present, it has a Young's modulus of approximately 1 TPa making it stronger than steel [7,8]. A variety of fields including biology [9–11] and medicine have begun tapping into the immense potential that graphene presents. For example, He *et al.* [12] used graphene as a surface-enhanced Raman scattering (SERS) substrate in order to carry out multiplex DNA detection. Rastogi *et al.* [13] discovered that cell adhesion and proliferation for neuronal and non-neuronal cells were enhanced by the use of single-layer graphene.

Many methods have been developed to fabricate graphene since the last decade. The prominent ones can be broadly classified into—(i) epitaxial growth of graphene [14], (ii) micromechanical exfoliation [15], (iii) exfoliation using electrochemical methods [16], (iv) exfoliation using solvents [17], and (v) chemical vapour deposition [18]. The fabrication of single-layer graphene is difficult and often requires expensive equipment. However, it has been observed that few-layer graphene (FLG) possesses certain properties that are similar to monolayer graphene such as the absence of gap in its electronic band structure [19] and its high surface area [20]. Due to these similarities, FLGs can substitute for single-layer graphene in various applications, generating cost-effective solutions.

Exfoliation using mechanical methods has generally helped develop high-quality graphene. Nevertheless, there is still a lot of work to be done in order to improve the efficiency of the process [1]. The exfoliation of graphene layers from bulk graphite is dependent on the Van der Waals forces between individual graphene layers being overcome through various means. Mechanical means to weaken these forces can be implemented through the application of shear or normal forces [21,22] leading to the graphite/graphene flakes being broken into smaller sizes. Although smaller flakes possess weaker Van der Waal forces, it prevents acquiring graphene sheets with the desirable large surface area [23].

Highly ordered pyrolytic graphite (HOPG) was used to produce graphene via micromechanical cleavage [22,24,25]. The normal force was applied on the surface of HOPG simply using Scotch tape, and after many iterations, single-layer graphene was produced. This procedure led to the discovery of monolayer graphene [21]. However, this technique can only be used to produce graphene in minute quantities [21]. Hernandez *et al.* produced graphene by sonicating graphite in organic solvents including N-N-dimethylformamide (DMF) and N-methylpyrrolidone (NMP), and the material was centrifuged after sonication. The limitation of this facile technique was that the graphene yield obtained turned out to be merely 0.01 mg ml$^{-1}$ [17]. Buzaglo *et al.* [26] devised a sonication method wherein nearly 100% of the graphite sheets were exfoliated and showed that their technique could be extended to ball milling and shear mixing. However, sonication works through the principle of cavitation assisting exfoliation leading to excessive local heat generation which causes the material being sonicated to be subjected to enormous pressure and sharp temperature changes [27–29] making it highly restrictive for industrial applications.

The principle behind the fabrication of graphene using ball milling is that the application of shear force on graphite by the milling balls leads to its exfoliation (figure 1*a,b*). This method yields graphene flakes of large size. However, during the milling process, it is probable that the balls may strike the graphite/graphene flakes normal to the surface causing them to reduce in lateral dimensions. Moreover, the crystal structure may also be adversely affected leading to the formation of amorphous material [23]. The ball milling apparatus usually employed to exfoliate graphite are stirred media mills [23] and planetary ball mills [30]. Stirring media mill enables better control of the heat generated during milling. Planetary ball mills simultaneously assist in functionalizing and exfoliating the material being milled due to the application of high energy during operation. The downside of this process is that it is time consuming and despite that the material might require further sonication [30].

Chen *et al.* produced hydrophilic graphene dispersions by ball milling graphite for 4 h at 400 r.p.m. along with poly(vinylpyrrolidone) and supercritical $CO_2$, using a stirring ball milling machine [31]. Supercritical materials are maintained above its critical pressure and temperature and have the ability to infiltrate gaseous substances and dissolve liquid substances [32]. $CO_2$ greatly assists in intercalation for it possesses a molecule size of 0.33 nm, comparable to the graphene interlayer

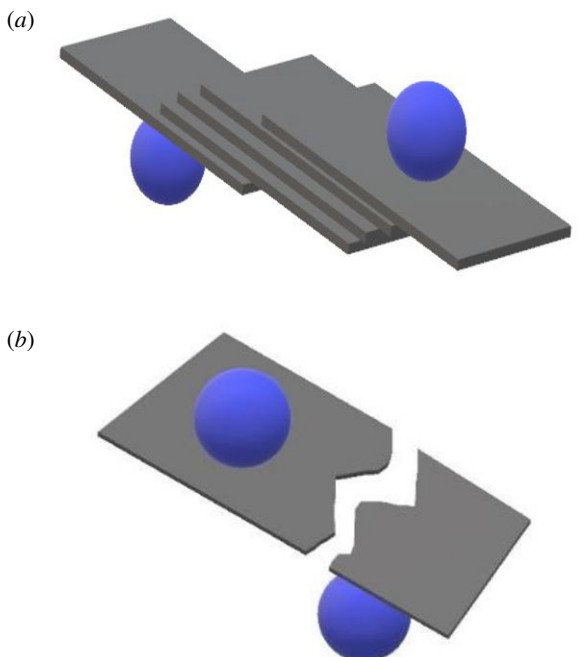

**Figure 1.** (*a*) Shear exfoliation of graphene layers. (*b*) Destruction of graphene planes due to normal impact.

distance of 0.34 nm [33]. Furthermore, $CO_2$ requires relatively low pressure and temperature to reach a critical point [32]. Similarly, Zhu *et al.* produced graphene by ball milling graphite with dry ice (solid $CO_2$) in a planetary mill for 24 h. The resulting powder was sonicated for 20 min in the presence of 1 M aqueous HCl solution leading to the formation of edge-carboxylated graphene (ECG). Thermally reduced graphene (TRG) was obtained by annealing the ECG at a temperature of 900°C for a time duration of 10 min [34]. Al-Sherbini *et al.* exfoliated graphene using 2-ethylhexanol and kerosene as solvents. A planetary ball mill was employed to conduct the high-energy milling operation for a duration of 60 h at 400 r.p.m. Centrifugation along with heat treatment was performed in the presence of Argon at 600°C [35]. These techniques although effective in exfoliating graphite, consisted of multiple steps. Zhao *et al.* conducted ball milling to exfoliate multi-layer graphite nanosheets into graphene. The solvent used was N,N-dimethylformamide (DMF) [36]. In this process, the graphitic material was already very thin, with thickness ranging between 30 and 80 nm. Additionally, DMF has been found to be very hazardous to living organisms [37]. Leon *et al.* developed few-layer graphene by milling graphite along with melamine using a planetary ball mill. The parameters for the graphene samples produced, speed of rotation and milling time, ranged between 100–250 r.p.m. and 30–60 min, respectively. The experiments conducted were either in the presence of nitrogen or a normal atmosphere [38]. Although a facile procedure was presented, the quantity of input material equalling 30 mg was relatively low. Alongside, González *et al.* produced graphene using a similar process, ball milling it in the presence of melamine at 100 r.p.m. for 30 min. The drawback was that their process took a time span of 6–7 days, for excess melamine had to be removed from graphene to meet safety constraints [39,40].

Lately, the influence of biological materials as exfoliates has also been investigated [41–44]. González *et al.* milled graphite at 250 r.p.m. in the presence of carbohydrate to produce graphene dispersions. They found that glucose showed the highest efficacy in exfoliating graphene with a reduced presence of defects [45]. Ahadian *et al.* [46] developed graphene dispersions by sonicating graphite in bovine serum albumin (BSA) media. BSA is a protein that is obtained from cows through natural means. BSA possesses both hydrophobic as well as hydrophilic sections. The hydrophobic section is adsorbed on graphene, which also is hydrophobic. This assists in the formation of dispersions and potentially prevents restacking of graphene [46]. Pattammattel *et al.* produced graphene using a kitchen blender after applying shear/turbulence force on graphite in the presence of BSA. They also explored the relation between varying BSA–graphite ratios on the exfoliation rate and came to the conclusion that the best results could be obtained from a ratio of 0.03 [47].

Ahadian *et al.* [46] established the molecular interaction occurring between BSA and graphene under the assumption that there is no interplay between graphene and the hydrophilic amino acid portion of

BSA. In order to better understand the underlying process, density functional theory (DFT) calculations of the charge transfer, binding energy and density of states (DOS) analyses were performed by them. The impact of individual hydrophobic amino acids in the exfoliation of graphene sheets was discerned by calculating the product of the number of amino acids with their respective binding energy values. The impact of the amino acids in descending order is, Leu, Pro, Ala, Cys, Val, Phe, Tyr, Gly, ILeu, Met and Tyrp. The charge transfer calculations helped determine that the removal of electrons from graphene was due to the effect of electronegative atoms from amino acid types such as imino, aliphatic and those possessing sulfur. The interaction between graphene and aromatic acids was mainly due to aromatic rings that were in plane to the graphene sheet. The DOS analysis showed that the interaction between graphene sheets and amino acids are non-covalent in nature. The dispersion of graphene sheets in water was due to the non-covalent interaction of BSA and graphene, which in turn was because of the effect of hydrophobic amino acids containing aromatic rings and aliphatic side chains [46].

This paper describes a green one-step technique to fabricate biocompatible graphene using a planetary ball mill. One of the objectives of this experimental study was to investigate techniques to facilitate the fabrication of mass-produced graphene; therefore, the aqueous dispersion of FLG produced is not centrifuged after it has been collected. Moreover, to use the beneficial effects that BSA lends to FLG's biocompatibility, these two materials are not separated in any post-processing step.

# 2. Material and method

BSA (CAS: 9048-46-8) and graphite (powder, less than 20 µm, synthetic, CAS: 7782-42-5) were purchased from Sigma Aldrich USA. The concentration of graphite was kept constant at 100 mg in each experiment throughout the study. The concentration of BSA was varied according to the design of the experiment. In addition, 5 ml of water was added to the graphite–BSA mixture to prepare a solution. A two-station horizontal planetary mill (Fritsch, Pulverisette 7) was used for the mechanochemical processing. The milling containers were made of 316L grade stainless steel. The speed of rotation was kept constant at 300 r.p.m., with the milling jars being kept at room temperature. The direction of rotation alternated along with intermittent pauses to prevent overheating. Sixteen chrome steel balls (AISI E52100, $\rho \sim 7.83$ g cm$^{-3}$), each weighing approximately 8.3 g were placed in each of the steel jars.

# 3. Results and discussion

The texture of the fluid recovered after ball milling varied depending upon the amount of BSA present. FLG fluids in the presence of minute quantities of BSA (less than or equal to 10%) had smooth textures. Whereas FLG that was exfoliated in the presence of large amounts of BSA (greater than or equal to 50%) displayed a foamy texture and usually had to be scooped out using a spatula (electronic supplementary material, figure S1a and S1b). Additionally, the dispersions with large concentrations of BSA were found to disperse for several days

## 3.1. Effect of bovine serum albumin concentration and milling time on exfoliation

X-ray diffraction was primarily used to investigate the exfoliation of graphene. The reduction in the size of graphite flakes is caused due to the diminishing of the *n-n* stacked layers and is indicated by the decreasing of the (002) peak intensity. Furthermore, this demonstrates that the size of the graphite particles is reducing normal to the basal plane. Also, there is an increase in the 002 peak's full width at half maxima (FWHM) due to Scherrer broadening along with the reduction in peak intensity [48,49].

While ball milling time was a contributor to the exfoliation of graphene sheets, it was found that ball milling for shorter durations of time produced sharp $2\theta$ peaks (figure 2a–d). This could be attributed to the fact that graphite particles are randomly oriented which results in the flakes experiencing a combination of compressive and shear forces, which might cancel each other.

Additionally, the individual graphene layers generally restack unless there is an intercalation agent preventing it [48,50]. It was noted that after ball milling graphite for 90 h in the absence of BSA, the $2\theta$ peaks produced were broad and graphene like (figure 3a–d). Also, this is the first time that graphene has been exfoliated in the absence of any intercalating agent by ball milling. However, the application of such prolonged times can be considered uncompetitive for industrial applications. Also, samples milled for long periods of time can get contaminated, especially from materials present in the

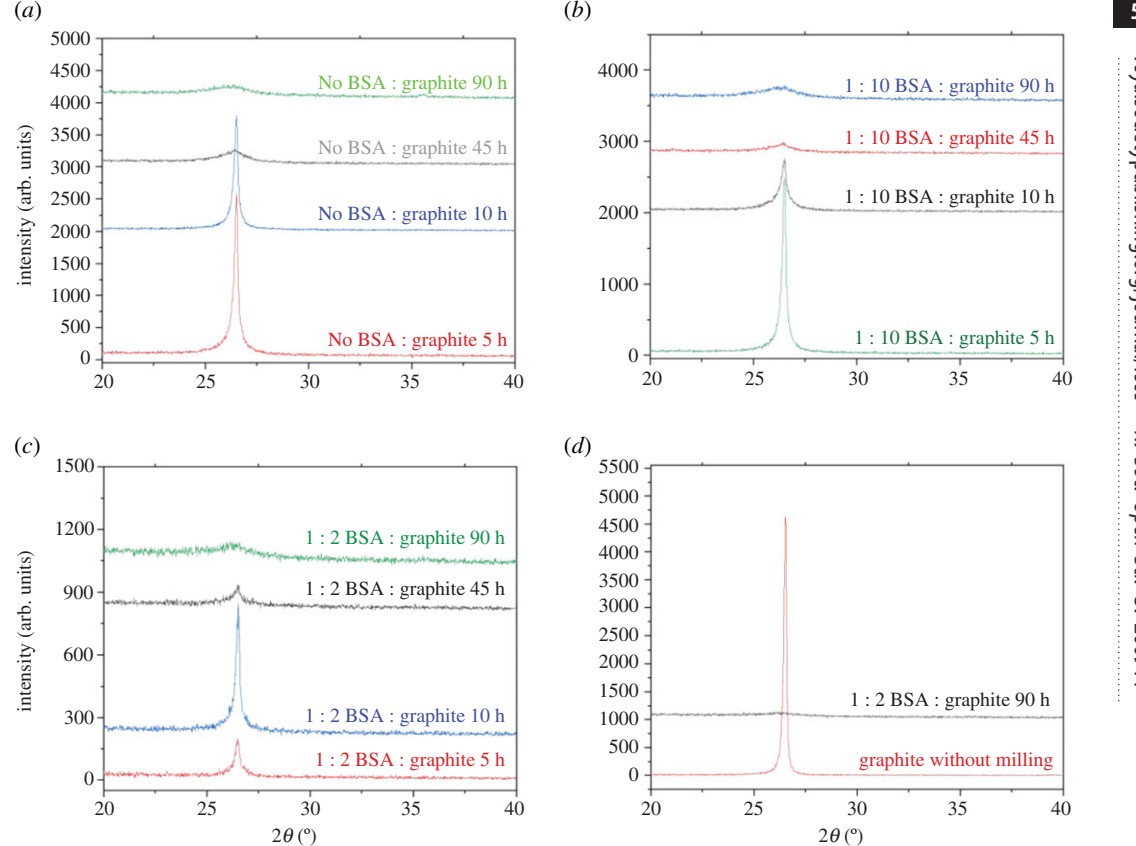

**Figure 2.** (a) Evolution of 002 Bragg peak for varying time periods after (a) ball milling graphite in the absence of BSA, (b) ball milling graphite and BSA in the ratio of 1 : 10 and (c) ball milling graphite and BSA in the ratio of 1 : 2. (d) Ball-milled graphene and BSA in the ratio of 1 : 2, compared with pure graphite.

balls as well as the milling jar. It was observed that the graphene produced after milling for 90 h, without BSA, was magnetic, which was attributed to iron contamination from the steel jars.

Moreover, increasing the initial concentration of BSA helped expedite the separation of graphene sheets. The effect of BSA in the exfoliation of graphene is prominent for shorter milling times, although its effects are subtly visible for longer periods. Furthermore, XRD of pure BSA and 45 h milled BSA was performed to analyse whether milling led to the formation of amorphous BSA. One characteristic XRD peak of BSA disappeared after the milling process, making it difficult to reach a conclusion of the nature of the material after milling was performed (electronic supplementary material, figure S2).

## 3.2. Optimization of milling parameters

The initial experimental design of this study was to investigate the effects of specific ball milling times with the increase in the initial concentration of BSA. However, after noting the impactful effects of BSA in the exfoliation of graphene, the milling time was drastically cut short to further optimize the process. Milling equal concentrations of BSA and graphite for an hour gave significantly better results than those obtained after 10 h in the absence of BSA (figure 4a,b).

## 3.3. Scanning electron microscopy results

For purity critical applications, it is imperative that the synthesized graphene be investigated for materials that are present in the milling jar after long mechanochemical operations. Therefore, energy-dispersive X-ray spectroscopy (EDS) was performed to gauge the level of contamination in the synthesized sample, milled for 45 h (figure 5a,b). Multiple runs (spectrum 1 and 2) were performed on different samples to obtain a fair estimate of the contamination. The sample contains approximately 3.5 atomic % impurity consisting of mainly Fe and Cr from the milling vials as shown

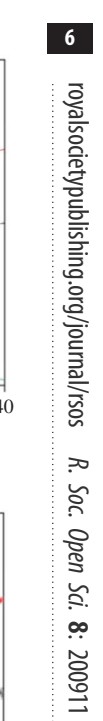

**Figure 3.** Evolution of 002 Bragg peak for varying concentrations of BSA after (*a*) 5 h of ball milling, (*b*) 10 h of ball milling, (*c*) 45 h of ball milling and (*d*) 90 h of ball milling.

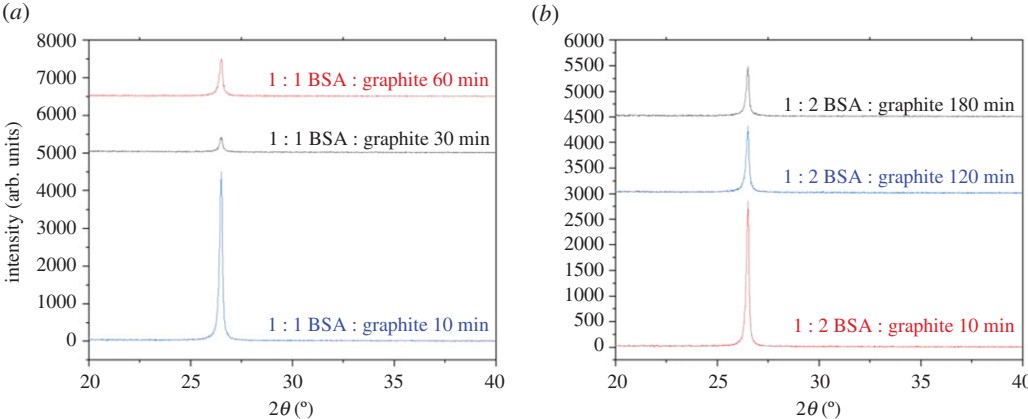

**Figure 4.** (*a*) Evolution of 002 Bragg peak for (*a*) 1 : 1 BSA : graphite after milling for short time periods, (*b*) 1 : 2 BSA : graphite after milling for short time periods.

in table 1. The EDS data were also carefully analysed for other impurities and were found to be insignificant.

## 3.4. Transmission electron microscopy results

Transmission electron microscopy (TEM) was employed to further discern the structure of the ball-milled material. The FLG was in the form of folded nanosheets; however, the sample containing BSA was found to contain monolayer graphene. Moreover, it can be verified that despite the same number of milling

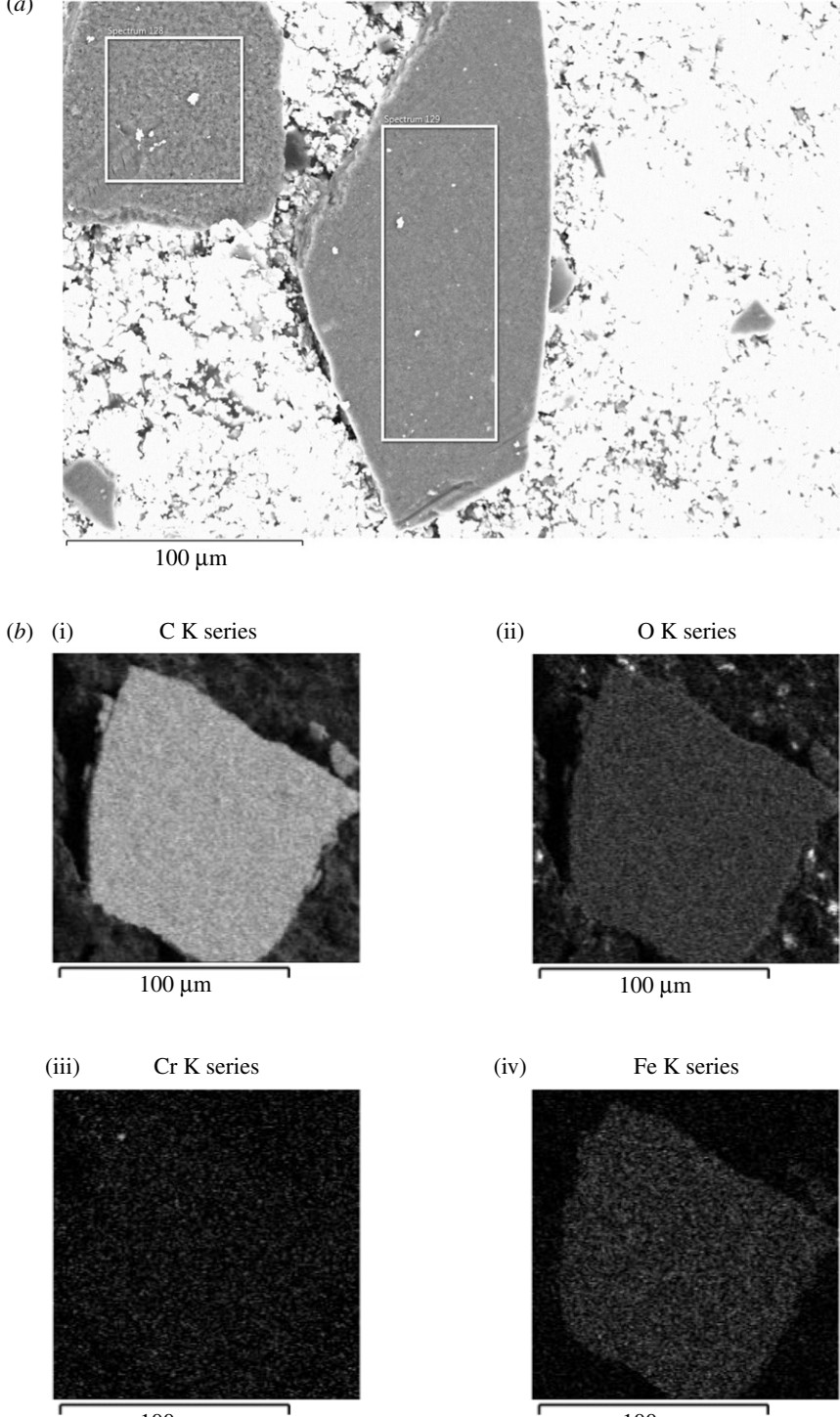

**Figure 5.** (*a*) Energy-dispersive X-ray spectroscopy of synthesized graphene (1 : 2 BSA : graphite 45 h, square: spectrum 1, rectangle: spectrum 2). (*b*) Elemental analysis of synthesized graphene (1 : 2 BSA : graphite 45 h).

**Table 1.** EDS characterization of synthesized sample (1 : 2 BSA : graphite 45 h, at%).

| spectrum label | C | Cr | Fe |
|---|---|---|---|
| spectrum 1 | 96.59 | 0.38 | 3.03 |
| spectrum 2 | 96.47 | 0.36 | 3.18 |

(*a*)

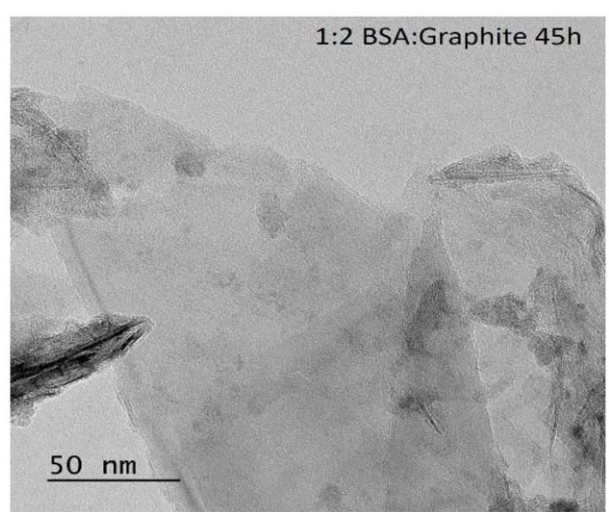

(*b*)

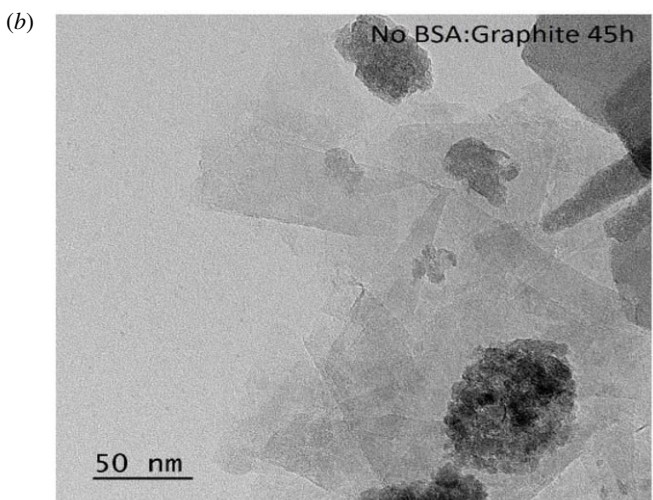

**Figure 6.** (*a*) Transmission electron microscope image of (*a*) 45 h milled 1 : 2 BSA–graphite sample and (*b*) 45 h milled graphite sample (no BSA).

hours, BSA-FLGs have a lower number of layers than those without BSA (figure 6*a*,*b*). Further TEM images at different magnifications have been included as supporting information (electronic supplementary material, figures S3 and S4).

## 3.5. Detection of disorder produced during the ball milling process

The effect of ball milling on the production of defects and graphene exfoliation was determined using Raman spectroscopy. The D and G peaks are the prominent peaks observed in the Raman spectra of carbon materials and other polyaromatic hydrocarbons [51,52]. These peaks can be observed at approximately 1360 and 1560 cm$^{-1}$ [51,53,54]. The appearance of the D peak is an indicator of defects within the material [44–46]. Ferrari *et al*. collected Raman spectra using 514.5 nm laser excitation on graphite and graphene. The G peak and the 2D band were acutely observed at approximately 1580 and 2700 cm$^{-1}$, respectively [55,56]. An additional peak was observed in the spectra at approximately 3250 cm$^{-1}$ and has been referred to as the 2D′ peak [53,55].

The defects produced due to the milling process are overall directly proportional to the number of hours the process was carried out. However, no clear trend could be discerned to describe the effect of BSA on the formation of defects. The $I_D/I_G$, a numerical value denoting the production of defects, ranged from 0.48 to 1.19, the latter arising after grinding was performed for 90 h (figure 7*a*,*b*). González *et al*. obtained an $I_D/I_G$ of approximately 1.8 after milling for 8 h using glucose. However, even after milling for 90 h using BSA, the BSA-FLG has lower defect ratios than that produced using

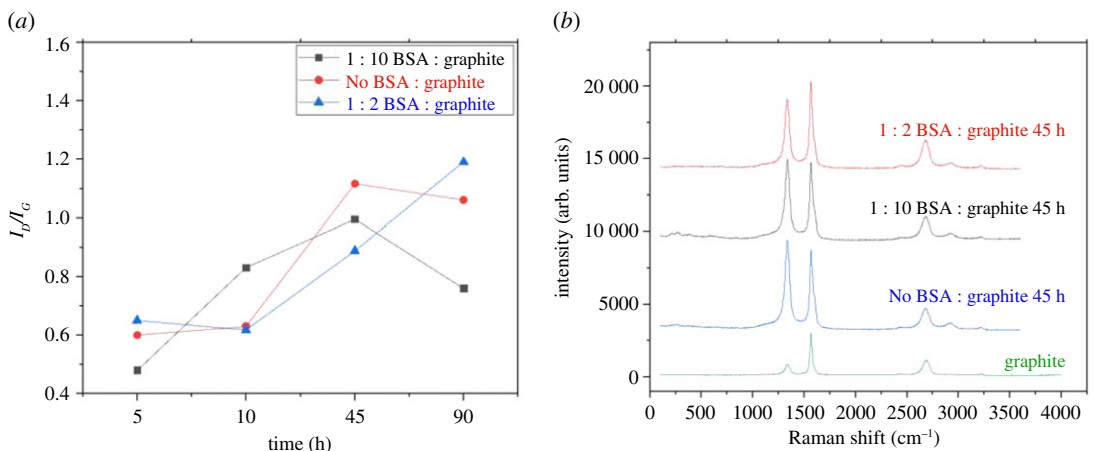

**Figure 7.** (a) Variation in the $I_D/I_G$ ratio with milling time and concentration of BSA. (b) Raman spectra for graphite and graphene after 45 h of milling.

glucose as an exfoliating agent [45]. Additionally, Pattammattel *et al.* discovered that the $I_D/I_G$ ratios reached up to 0.6 after exfoliating graphene in the presence of BSA using the shearing force applied by a kitchen blender. It can be inferred that any application of shear force for the production of graphene tends to generate defects in the material [47].

The nanographite size ($L_a$) has usually been characterized using $I_D/I_G$. The qualitative control of structural transformation in various graphitic materials can be calibrated by Tuinstra and Koenig's (T-K) empirical relation [57]. The amount of initial defects is given by $L_a^{-1}$ and referred to as the defect density $n_D$. $L_a$ is proportional to the quantity of disorder in nano-crystallites. Furthermore, the defect distance in graphene containing zero-dimensional point defects can also be obtained through the correlations given below.

$$L_D^2 \ (\text{nm}^2) = \frac{(4.3 \pm 1.3)*10^3}{E_L^4} \left(\frac{I_D}{I_G}\right)^{-1}$$

and

$$n_{D(\text{cm}^{-2})} = (7.3 \pm 2.2)*10^9 E_L^4 \left(\frac{I_D}{I_G}\right).$$

The above correlations can only be used under the assumption that graphene samples having point defects at distances satisfying $L_D \geq 10$ nm observed through excitation light within the visible range [58,59].

Based on the two equations given above and taking the laser energy to be 2.33 eV [60], our BSA-FLG samples with $I_D/I_G = 0.48$ contain $L_D$ ranging between 19.89 and 14.56 nm. The $n_D$ ranges between $7.21 \times 10^{10}$ and $1.34 \times 10^{11}$ cm$^{-2}$. We have not shown the $L_D$ and $n_D$ for the 90 h milled samples since its $L_D < 10$ nm, violating the above assumptions.

The FWHM increases with milling and the addition of BSA, which is similar to the results obtained by Pattammattel *et al.* [47].

## 3.6. Integration of bovine serum albumin few-layer graphene with astrocyte cells

Astrocytes are a type of glial cell abundantly found in the central nervous system (CNS). They perform a variety of functions including axon guidance, synaptic support, control blood–brain barrier and are very responsive to CNS attacks [61]. Sasidharan *et al.* discovered that cell apoptosis occurs due to pristine graphene collecting around its membrane. This was attributed to the powerful hydrophobic interaction that pristine graphene had with the hydrophobic cellular membrane. They theorized that essential nutrients, proteins and ion channels were sealed off by pristine graphene, which leads to the formation of reactive oxygen species (ROS) stress. However, they found that hydrophilic graphene did not exhibit the harmful traits that hydrophobic graphene possessed. They concluded that surface functionalization is one of the salient factors that mitigate the cytotoxic characteristics of pristine graphene [62–64]. Since the hydrophobic portion of BSA is adsorbed on graphene, the hydrophilic part interacts with water and the surrounding environment promoting a benign interaction with cells.

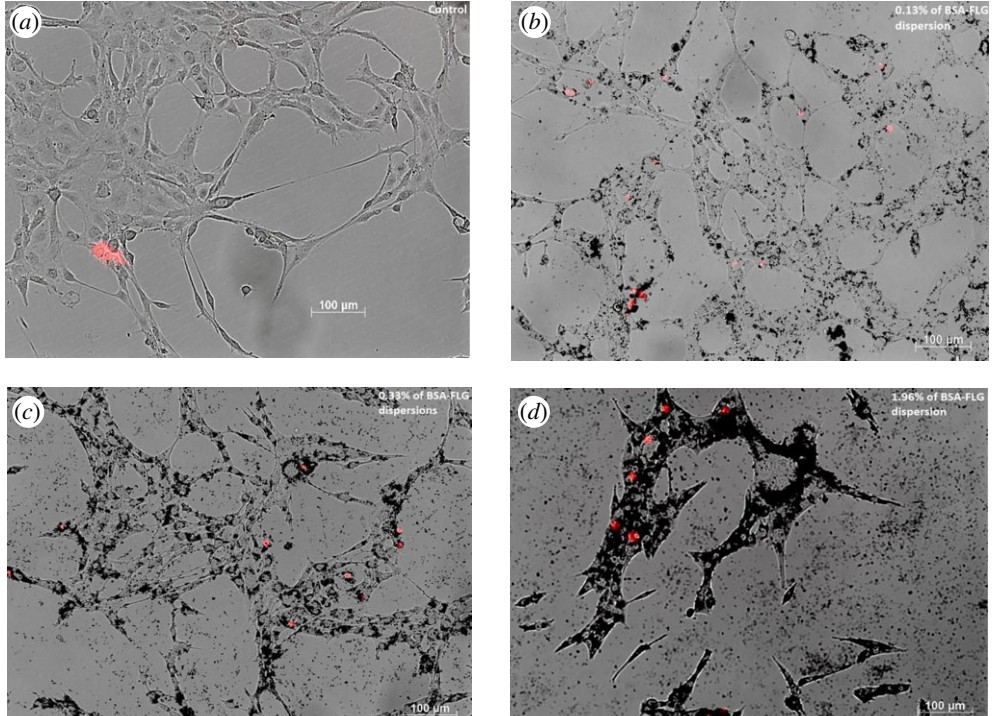

**Figure 8.** Inverted microscope image of (*a*) astrocyte cells (control), (*b*) astrocyte cells integrated with 0.13% of BSA-FLG dispersion, (*c*) astrocyte cells integrated with 0.33% of BSA-FLG dispersion and (*d*) astrocyte cells integrated with 1.96% of BSA-FLG dispersion. Red dots indicate cell death.

BSA-FLG was placed in cell media with astrocytes cells (ATCC) at varying concentrations, for a duration of approximately 3 days to experimentally study its effects on them. Propidium iodide (PI) dye, which detects cell viability, was used to identify cell death. It does not penetrate live cells containing intact membranes; however, it does enter damaged/dead cells and intercalates with the RNA and DNA bases, thereby binding to them. The control contained only cells and no graphene. The optical imaging technique employed shows that at lower concentrations of BSA-FLG dispersions (approx. 0.13%) only a small proportion of cells were affected at the locations studied. However, at higher concentrations (1.96%), a larger number of cells died. PI dye was used to identify the dead cells and they have been marked in red (figure 8*a*–*d*).

# 4. Conclusion

This paper investigates a mechanochemical process to fabricate biocompatible graphene. The effect of BSA on the accelerated exfoliation of graphene while producing relatively lower defects can be easily discerned. Additionally, this paper shows the production of graphene in the absence of any exfoliating agent after milling for large time periods. Preliminary investigations found that lower concentrations of BSA-functionalized graphene can be integrated with cells without inducing their death. Future work can include investigations of the optimal amount of BSA that allow high concentrations of graphene to be placed within the cell media. This approach may be applied in areas that require a cheap one-step fabrication method for FLG while recognizing the impact of milling speed and BSA–graphite concentrations.

# 5. Experimental

## 5.1. Powder X-ray diffraction analysis

The powder X-ray diffraction (PXRD) analysis was performed at room temperature using a PANalytical X'PERT diffractometer. Cu-K$_{\alpha 1}$ radiation was employed with a 0.02° 2$\theta$ step, in the 2$\theta$ range from 10° to 80°. In order to prevent the characterized sample from being exposed to oxygen and moisture, a polyimide (Kapton) film was implemented. This causes the generation of amorphous-like background

at $13° \leq 2\theta \leq 20°$. The powders of FLG were obtained by drying the suspension in an oven kept at 60°C. The as-dried powder was fixed on the zero-background silicon substrate using vacuum grease.

## 5.2. Scanning electron microscopy

The SEM images were taken using FEI Teneo Lovac FE-SEM, using an accelerating voltage of 10 kV and reported beam current of 1.6 nA. The powders of FLG were obtained by drying the suspension in an oven kept at 60°C. The as-dried powder was sprinkled sparingly on conductive carbon tape.

## 5.3. Energy-dispersive X-ray spectroscopy

The EDS analysis was performed using an Oxford Aztec system with X-Max 80 detector, attached to the Teneo.

## 5.4. Transmission electron microscopy

The TEM images were taken using a JEOL 2100 scanning and electron microscope. A Gatan OneView 4 K camera was used to capture images. The sample preparation included pipetting 2 µl of the aqueous FLG dispersion onto a carbon film copper grid. The excess material was removed using a filter paper and a thin film of the material was used for characterization.

## 5.5. Cell imaging

The pictures of cells were captured using an inverted microscope platform (overall view Axio Observer.A1). The samples were prepared using BSA-FLG at varying concentrations. Propidium iodide staining was conducted to identify cell death.

## 5.6. Raman spectroscopy

Raman spectra were collected using a Horiba XploRa Plus confocal Raman upright microscope equipped with a 532 nm excitation source (1.5 mW at the sample) and a Synapse EMCCD camera. A $50 \times$ air objective (Olympus, LMPlanFL) with 0.25 numerical aperture was used to collect Raman spectra in the epi-direction. The spectra were collected from 600–3300 cm$^{-1}$ with a 600 grooves mm$^{-1}$ grating, each spectrum corresponds to an average of three measurements with a 30 s acquisition time and two accumulations. Sample preparation involved putting a drop on a silicon wafer and letting it dry overnight.

Data accessibility. Our data are deposited at Dryad Digital Repository: http://doi.org/10.5061/dryad.sqv9s4n1n [65].

Authors' contributions. D.-G.T. carried out the laboratory work, participated in data analysis, participated in the design of the study and drafted the manuscript; S.D.-A. participated in data analysis and critically revised the manuscript; S.G. participated in data analysis and critically revised the manuscript; V.K.P. and D.M.-P. participated in data analysis; R.M. conceived of the study, acquired funding and critically revised the manuscript; E.A.S. helped draft the manuscript; N.N.H. and R.M. conceived of the study, acquired funding and critically revised the manuscript. All authors gave final approval for publication and agree to be held accountable for the work performed therein.

Competing interests. We declare we have no competing interests.

Funding. This work was partially supported by the Office of Naval Research grant no. N000141712620, US Department of Energy grant no. DE-AC02-07CH11358 and Army Research Office grant no. W911NF1710584.

Acknowledgements. The authors would like to thank Marilyn McNamara, Alex Wrede and Jingshuai Guo for their assistance in the cell culture preparations and characterization using confocal imaging. The Raman measurements were supported by the US Department of Energy, Office of Basic Energy Sciences, Division of Chemical Sciences, Geosciences, and Biosciences through the Ames Laboratory. The Ames Laboratory is operated for the US Department of Energy by Iowa State University under Contract No. DE-AC02-07CH11358.

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
