## [Peer Review File · Royal Society Open Science]

Review History

RSOS-200911.R0 (Original submission)

Review form: Reviewer 1

Is the manuscript scientifically sound in its present form?

Yes

Are the interpretations and conclusions justified by the results?

Yes

Is the language acceptable?

Yes

Do you have any ethical concerns with this paper?

No

Have you any concerns about statistical analyses in this paper?

No

Recommendation?

Major revision is needed (please make suggestions in comments)

Comments to the Author(s)

The authors present a method to produce few layer graphene using ball milling in presence of bovine serum albumin (BSA). This method for graphene exfoliation from graphite have already been presented in multiple reports using single proteins or full serum, and there are significant contributions to the field that were not cited in this manuscript (Laaksonen, P. et al. *Angew. Chem. Int. Ed.* 49, 4946–4949 (2010), Castagnola, V., et al. *Nature communications* 9.1 (2018): 1-9, Gravagnuolo, A. M. et al. *Adv. Funct. Mater.* 25, 2771–2779 (2015), Paredes, J. I. & Villar-Rodil, S. *Nanoscale* 8, 15389–15413(2016)).

The fact that BSA help stabilizing the flakes produced by the milling exfoliation is very well known therefore the reviewer feels it is difficult to find elements of novelty or insight in this manuscript. The authors should highlight more the specific rationale of the study, the increment over the published works and better link the results obtained to the application.

The material could have been characterized more in terms of yield, lateral size, overtime stability etc., trying to correlate these features to the milling conditions (absence/presence of BSA, BSA/graphite ratio, milling time, etc).

Also, the material is called biocompatible but the biological investigation is quite poor. In Fig. 8 only one optical microscope image is shown for each condition. The use of PI to detect cell necrosis is mentioned in materials but the results are not shown.

The choice of astrocytes over other cell lines is not explained. The author could improve this part by adding a quantification of the cell viability with some assay. Another important aspect is the influence of BSA (that is probably highly aggregated or unfolded after the long treatment) and the influence of the contaminants on the viability.

The author should use appropriate controls to investigate these aspects.

Minor comments:

- Red dots in Fig 8 are not described
- Pag. 13 line 40 : problem with references
- Starting graphite spectrum in Fig 7b show high presence of defects. The reference for the commercial starting material is missing.

Review form: Reviewer 2

Is the manuscript scientifically sound in its present form?

Yes

Are the interpretations and conclusions justified by the results?

No

Is the language acceptable?

Yes

Do you have any ethical concerns with this paper?

No

Have you any concerns about statistical analyses in this paper?

No

Recommendation?

Major revision is needed (please make suggestions in comments)

Comments to the Author(s)

The paper investigates exfoliation of graphite by ball milling in the presence of BSA as “green” stabilizer. The resultant material is characterised by XRD, TEM, SEM/EDX and Raman spectroscopy as function of milling time and BSA concentration. Qualitatively, it was found that relatively short milling times can be used for the exfoliation which avoids contamination of the graphene with the milling media. The results are interesting and the paper is well written. The introduction is relatively lengthy, but provides a good overview of the literature. Acceptance after the following revisions is suggested:

- 1) The main issue is that the observations are all rather qualitative in nature. The authors are encouraged to make a bit more quantitative analysis.
 - i) The dispersed concentration could be estimated based on UVVis spectroscopy. While the protein will also give signal (see for example SI of Nature Commun. 2018, 9 (1), 1577), the extinction at long wavelength can be used to analyse the dispersed concentration as function of time
 - ii) The Raman spectra should be investigated in more detail. For example, the authors are encouraged to analyse D/G ratio and G band width which allows to draw (semi-quantitative) conclusions on edge and basal plane defectiveness (2D Mater. 2017, 4 (2), 025039). In addition the 2D band shape could be analysed to get a more quantitative picture on degree of exfoliation (at least as comparison); example procedures are described in various articles (Carbon 2015, 81 (0), 284-294, Nanoscale 2016, 8 (7), 4311-4323., Carbon 2013, 53, 357-365. Carbon 2020, 161, 181-189.). While all these methods have some restrictions, they would allow for a more solid sample comparison.
- 2) The plot of Raman D/G ratio as function of time in Figure 7a shows a significant amount of scatter. Is this related to spot to spot variations in each sample from the three measurements that were averaged? The authors are encouraged to average over more spots and include the standard deviation in the figure. For Figure 7b, they have chosen to show the data at 45h which follows a decreasing D/G ratio in the order no BSA, 1:10 BSA and 1:2 BSA. However, the plot above shows this is more or less a coincidence which varies for the different times. As such, it is not representative and it cannot be concluded that the D/G ratio is lower in the presence of BSA (as currently stated)
- 3) The sample preparation for each characterization should be included in the experimental section.

Decision letter (RSOS-200911.R0)

Dear Miss Hashemi:

Title: PROTEIN ASSISTED SCALABLE MECHANOCHEMICAL EXFOLIATION OF FEW LAYER BIOCOMPATIBLE GRAPHENE NANOSHEETS
Manuscript ID: RSOS-200911

The editor assigned to your manuscript has now received comments from reviewers. We would like you to revise your paper in accordance with the referee and Subject Editor suggestions which can be found below (not including confidential reports to the Editor). Please note this decision does not guarantee eventual acceptance.

Please submit your revised paper before 01-Jan-2021. Please note that the revision deadline will expire at 00.00am on this date. If we do not hear from you within this time then it will be assumed that the paper has been withdrawn. In exceptional circumstances, extensions may be possible if agreed with the Editorial Office in advance. We do not allow multiple rounds of revision so we urge you to make every effort to fully address all of the comments at this stage. If deemed necessary by the Editors, your manuscript will be sent back to one or more of the original reviewers for assessment. If the original reviewers are not available we may invite new reviewers.

Please also include the following statements alongside the other end statements. As we cannot publish your manuscript without these end statements included, if you feel that a given heading is not relevant to your paper, please nevertheless include the heading and explicitly state that it is not relevant to your work.

- Ethics statement

Please clarify whether you received ethical approval from a local ethics committee to carry out your study. If so please include details of this, including the name of the committee that gave consent in a Research Ethics section after your main text. Please also clarify whether you received informed consent for the participants to participate in the study and state this in your Research Ethics section.

OR

Please clarify whether you obtained the necessary licences and approvals from your institutional animal ethics committee before conducting your research. Please provide details of these licences and approvals in an Animal Ethics section after your main text.

OR

Please clarify whether you obtained the appropriate permissions and licences to conduct the fieldwork detailed in your study. Please provide details of these in your methods section.

- Data accessibility

It is a condition of publication that you make available the data and research materials supporting the results in the article. Datasets should be deposited in an appropriate publicly available repository and details of the associated accession number, link or DOI to the datasets must be included in the Data Accessibility section of the article (<https://royalsocietypublishing.org/rsos/for-authors#question17>). Reference(s) to datasets should also be included in the reference list of the article with DOIs (where available).

Please include a Data Availability section after your main text stating where supporting data are available from, or where they will be made available should your article be accepted for publication.

If you wish to submit your supporting data or code to Dryad (<http://datadryad.org/>), or modify your current submission to dryad, please use the following link:
<http://datadryad.org/submit?journalID=RSOS&manu=RSOS-200911>

- **Competing interests**

Please include a Competing Interests section after your main text declaring any financial or non-financial competing interests. If you have no competing interests please state 'I/we have no competing interests.'

- **Authors' contributions**

Please include an Authors' Contributions section at the end of your main text detailing the contribution of each author. All authors should have read and approved the manuscript before submission and this should be stated in the Authors' Contributions section.

The list of Authors should meet all of the following criteria; 1) substantial contributions to conception and design, or acquisition of data, or analysis and interpretation of data; 2) drafting the article or revising it critically for important intellectual content; and 3) final approval of the version to be published.

- **Acknowledgements**

- **Funding statement**

Please include a funding section after your main text which lists the source of funding for each author.

Royal Society of Chemistry
Thomas Graham House
Science Park, Milton Road
Cambridge, CB4 0WF

Royal Society Open Science - Chemistry Editorial Office

On behalf of the Subject Editor Professor Anthony Stace and the Associate Editor Professor Tobias Hertel.

RSC Associate Editor: 1
Comments to the Author:
(There are no comments.)

RSC Associate Editor: 2
Comments to the Author:
(There are no comments.)

Reviewers' Comments to Author:
Reviewer: 1

Comments to the Author(s)

The authors present a method to produce few layer graphene using ball milling in presence of bovine serum albumin (BSA). This method for graphene exfoliation from graphite have already been presented in multiple reports using single proteins or full serum, and there are significant contributions to the field that were not cited in this manuscript (Laaksonen, P. et al. *Angew. Chem. Int. Ed.* 49, 4946-4949 (2010), Castagnola, V., et al. *Nature communications* 9.1 (2018): 1-9, Gravagnuolo, A. M. et al. *Adv. Funct. Mater.* 25, 2771-2779 (2015), Paredes, J. I. & Villar-Rodil, S. *Nanoscale* 8, 15389-15413(2016)).

The fact that BSA help stabilizing the flakes produced by the milling exfoliation is very well known therefore the reviewer feels it is difficult to find elements of novelty or insight in this manuscript. The authors should highlight more the specific rationale of the study, the increment over the published works and better link the results obtained to the application.

The material could have been characterized more in terms of yield, lateral size, overtime stability etc., trying to correlate these features to the milling conditions (absence/presence of BSA, BSA/graphite ratio, milling time, etc).

Also, the material is called biocompatible but the biological investigation is quite poor. In Fig. 8 only one optical microscope image is shown for each condition. The use of PI to detect cell necrosis is mentioned in materials but the results are not shown.

The choice of astrocytes over other cell lines is not explained. The author could improve this part by adding a quantification of the cell viability with some assay. Another important aspect is the influence of BSA (that is probably highly aggregated or unfolded after the long treatment) and the influence of the contaminants on the viability.

The author should use appropriate controls to investigate these aspects.

Minor comments:

- Red dots in Fig 8 are not described
- Pag. 13 line 40 : problem with references
- Starting graphite spectrum in Fig 7b show high presence of defects. The reference for the commercial starting material is missing.

Reviewer: 2

Comments to the Author(s)

The paper investigates exfoliation of graphite by ball milling in the presence of BSA as “green” stabilizer. The resultant material is characterised by XRD, TEM, SEM/EDX and Raman spectroscopy as function of milling time and BSA concentration. Qualitatively, it was found that relatively short milling times can be used for the exfoliation which avoids contamination of the graphene with the milling media. The results are interesting and the paper is well written. The introduction is relatively lengthy, but provides a good overview of the literature. Acceptance after the following revisions is suggested:

- 1) The main issue is that the observations are all rather qualitative in nature. The authors are encouraged to make a bit more quantitative analysis.
 - i) The dispersed concentration could be estimated based on UVVis spectroscopy. While the protein will also give signal (see for example SI of Nature Commun. 2018, 9 (1), 1577), the extinction at long wavelength can be used to analyse the dispersed concentration as function of time
 - ii) The Raman spectra should be investigated in more detail. For example, the authors are encouraged to analyse D/G ratio and G band width which allows to draw (semi-quantitative) conclusions on edge and basal plane defectiveness (2D Mater. 2017, 4 (2), 025039). In addition the 2D band shape could be analysed to get a more quantitative picture on degree of exfoliation (at least as comparison); example procedures are described in various articles (Carbon 2015, 81 (0), 284-294, Nanoscale 2016, 8 (7), [4311-4323](tel:4311-4323)., Carbon 2013, 53, 357-365. Carbon 2020, 161, 181-189.). While all these methods have some restrictions, they would allow for a more solid sample comparison.
- 2) The plot of Raman D/G ratio as function of time in Figure 7a shows a significant amount of scatter. Is this related to spot to spot variations in each sample from the three measurements that were averaged? The authors are encouraged to average over more spots and include the standard deviation in the figure. For Figure 7b, they have chosen to show the data at 45h which follows a decreasing D/G ratio in the order no BSA, 1:10 BSA and 1:2 BSA. However, the plot above shows this is more or less a coincidence which varies for the different times. As such, it is not representative and it cannot be concluded that the D/G ratio is lower in the presence of BSA (as currently stated)
- 3) The sample preparation for each characterization should be included in the experimental section.

Author's Response to Decision Letter for (RSOS-200911.R0)

See Appendix A.

RSOS-200911.R1 (Revision)

Review form: Reviewer 1

Is the manuscript scientifically sound in its present form?

Yes

Are the interpretations and conclusions justified by the results?

Yes

Is the language acceptable?

Yes

Do you have any ethical concerns with this paper?

No

Have you any concerns about statistical analyses in this paper?

No

Recommendation?

Accept as is

Comments to the Author(s)

The authors improved the manuscript adding some rationale for the investigation and the analysis of lateral size and defect density.

Review form: Reviewer 2

Is the manuscript scientifically sound in its present form?

Yes

Are the interpretations and conclusions justified by the results?

Yes

Is the language acceptable?

Yes

Do you have any ethical concerns with this paper?

No

Have you any concerns about statistical analyses in this paper?

No

Recommendation?

Accept with minor revision (please list in comments)

Comments to the Author(s)

The authors have addressed all comments made by the referees and revised their manuscript appropriately. Unfortunately, the additional experiments both reviewers asked for could not be performed which would have improved the quality of the manuscript significantly. Nonetheless, it contains a sufficient amount of novelty and publication is recommended.

Decision letter (RSOS-200911.R1)

Dear Miss Hashemi:

Title: PROTEIN ASSISTED SCALABLE MECHANOCHEMICAL EXFOLIATION OF FEW LAYER BIOCOMPATIBLE GRAPHENE NANOSHEETS
Manuscript ID: RSOS-200911.R1

It is a pleasure to accept your manuscript in its current form for publication in Royal Society Open Science. The chemistry content of Royal Society Open Science is published in collaboration with the Royal Society of Chemistry.

On behalf of the Subject Editor Professor Anthony Stace and the Associate Editor Professor Tobias Hertel.

RSC Associate Editor:
Comments to the Author:
(There are no comments.)

RSC Subject Editor:
Comments to the Author:
(There are no comments.)

Reviewer(s)' Comments to Author:

Reviewer: 2

Comments to the Author(s)

The authors have addressed all comments made by the referees and revised their manuscript appropriately. Unfortunately, the additional experiments both reviewers asked for could not be performed which would have improved the quality of the manuscript significantly. Nonetheless, it contains a sufficient amount of novelty and publication is recommended.

Reviewer: 1

Comments to the Author(s)

The authors improved the manuscript adding some rationale for the investigation and the analysis of lateral size and defect density.

Appendix A

Response to Reviewers

We are thankful to the reviewers for their time and consideration while making their thoughtful comments. We have done our utmost to work on the reviewers' comments and build our manuscript according to their expectations.

The reviewers' comments have been copied verbatim and an answer to their comments are provided directly below.

Reviewer 1:

1. This method for graphene exfoliation from graphite have already been presented in multiple reports using single proteins or full serum, and there are significant contributions to the field that were not cited in this manuscript (Laaksonen, P. et al. *Angew. Chem. Int. Ed.* 49, 4946–4949 (2010), Castagnola, V., et al. *Nature communications* 9.1 (2018): 1-9, Gravagnuolo, A. M. et al. *Adv. Funct. Mater.* 25, 2771–2779 (2015), Paredes, J. I. & Villar-Rodil, S. *Nanoscale* 8, 15389–15413(2016)).

The citations for the above papers have been added. Although, the comparison between our work and the above reports ^{1–4} presents common ground, our one-step method on fabricating bio-compatible few layer graphene (BSA-FLG), using a planetary mill, in the presence of BSA has not been replicated. In addition, most of the reports mentioned in this comment have performed exfoliation using ultrasonic exfoliation, whose pitfalls have been explained in the introduction of our paper. We have also included it here for the reviewer's convenience –

“However, sonication works through the principle of cavitation assisting exfoliation leading to excessive local heat generation which causes the material being sonicated to be subjected to enormous pressure and sharp temperature changes ^{5–7} making it highly restrictive for industrial applications.”

2. The fact that BSA help stabilizing the flakes produced by the milling exfoliation is very well known therefore the reviewer feels it is difficult to find elements of novelty or insight in this manuscript. The authors should highlight more the specific rationale of the study, the increment over the published works and better link the results obtained to the application

After reviewing the literature extensively, we have not found papers that apply planetary ball milling to produce BSA-FLG. Also, we focused on doing a systematic study of different parameters, i.e., BSA quantity and milling time to indicate the

quality of BSA-FLG formed through various forms of characterization. Moreover, we explained the pitfalls and advantages of extreme parameter values during the fabrication process.

We have modified our introduction to highlight the rationale of our research-

“This approach may be applied in areas that require a cheap one-step fabrication method for FLG while recognizing the impact of milling speed and BSA-graphite concentrations.”

3. The material could have been characterized more in terms of yield, lateral size, overtime stability etc., trying to correlate these features to the milling conditions (absence/presence of BSA, BSA/graphite ratio, milling time, etc).

In this manuscript, we focused on characterization of manufactured graphene in order to show the quality of graphene. We have shown Raman spectroscopy, TEM, and biocompatibility of graphene. We appreciate the comment and we agree with the reviewer on the future steps of this work in terms of yield and lateral size correlation to the milling.

Furthermore, we added the following analysis –

“The nanographite size (L_a) has usually been characterized using I_D/I_G . The qualitative control of structural transformation in various graphitic materials can be calibrated by Tuinstra and Koenig’s (T-K) empirical relation⁸. The amount of initial defects is given by L_a^{-1} and referred to as the defect density n_D . L_a is proportional to the quantity of disorder in nano crystallites. Furthermore, the defect distance in graphene containing 0-dimensional point defects can also be obtained through the correlations given below^{9,10}.

$$L_D^2 \text{ (nm}^2\text{)} = \frac{(4.3 \pm 1.3) * 10^3}{E_L^4} \left(\frac{I_D}{I_G}\right)^{-1}$$
$$n_{D(\text{cm}^{-2})} = (7.3 \pm 2.2) * 10^9 E_L^4 \left(\frac{I_D}{I_G}\right)$$

The above correlations can only be used under the assumption that graphene samples having point defects at distances satisfying $L_D \geq 10\text{nm}$ observed through excitation light within visible range¹⁰.

Based on the two equations given above and taking the laser energy to be 2.33 eV¹¹, our BSA-FLG samples with $\frac{I_D}{I_G} = 0.48$ contain L_D ranging between 19.89 nm

*and 14.56 nm. The n_D ranges between $7.21 * 10^{10} \text{ cm}^{-2}$ and $1.34 * 10^{11} \text{ cm}^{-2}$. We have not shown the L_D and n_D for the 90-hour milled samples since its $L_D < 10\text{nm}$, violating the above assumptions ”*

4. Also, the material is called biocompatible but the biological investigation is quite poor. In Fig. 8 only one optical microscope image is shown for each condition.

The work done on integrating BSA-FLG with Astrocytes was intended to be preliminary in nature in this specific manuscript, therefore limited resources were directed towards those experiments here. However, we have performed further investigations on biocompatibility of graphene in below papers:

1. M. C. McNamara, A. E. Niaraki Asli, J. Guo, J. Okuzono, R. Montazami and N. N. Hashemi, *Frontiers in Materials*, 2020, 7, 61.

2. J. Guo, A. E. Niaraki Asli, K. R. Williams, P. L. Lai, X. Wang, R. Montazami and N. N. Hashemi, *Biosensors*, 2019, 9, 112.

Both papers are available online for more information.

5. The use of PI to detect cell necrosis is mentioned in materials but the results are not shown.

We have modified the results section to explain the effect of PI – “Propidium Iodide (PI) dye, which detects cell viability, was used to identify cell death. It does not penetrate live cells containing intact membranes, however it does enter damaged/dead cells and intercalates with the RNA and DNA bases, thereby binding to them.”

“PI dye was used to identify the dead cells and they have been marked in red [Figure 8 a-d]”

6. The choice of astrocytes over other cell lines is not explained. The author could improve this part by adding a quantification of the cell viability with some assay. Another important aspect is the influence of BSA (that is probably highly aggregated or unfolded after the long treatment) and the influence of the contaminants on the viability. The author should use appropriate controls to investigate these aspects.

We thank the reviewer for this careful comment. We did not have a particular reason for choosing Astrocytes. Our group has been working with Astrocytes and we have encapsulated them in graphene fibrous structures as 3D cell culture

platforms. Further investigations on biocompatibility of our graphene is shown in papers: *Frontiers in Materials*, 2020, 7, 61 and *Biosensors*, 2019, 9, 112 from our group.

7. Red dots in Fig 8 are not described

We have made the correction both in the text as well as under the figures –

“PI dye was used to identify the dead cells and they have been marked in red [Figure 8 a-d]”

“Red dots indicate cell death.”

8. Pag. 13 line 40 : problem with references

The problem with references has been corrected.

9. Starting graphite spectrum in Fig 7b show high presence of defects. The reference for the commercial starting material is missing.

This reference was already present in the Materials and Method section of the manuscript – “Bovine Serum Albumin (CAS: 9048-46-8) and Graphite (powder, <20 µm, synthetic, CAS: 7782-42-5) were purchased from Sigma Aldrich USA.”

Reviewer 2:

1. 1) The main issue is that the observations are all rather qualitative in nature. The authors are encouraged to make a bit more quantitative analysis.

i) The dispersed concentration could be estimated based on UVVis spectroscopy. While the protein will also give signal (see for example *SI of Nature Commun.* 2018, 9 (1), 1577), the extinction at long wavelength can be used to analyse the dispersed concentration as function of time

In this manuscript, we focused on characterization of manufactured graphene in order to show the quality of graphene. We have shown Raman spectroscopy, TEM, X-ray powder diffraction analysis, and biocompatibility of graphene. Our group has also been working with Astrocyte cells and we have encapsulated them in graphene fibrous structures as 3D cell culture platforms. Further investigations on biocompatibility of our graphene is shown in below papers from our group.

1. M. C. McNamara, A. E. Niaraki Asli, J. Guo, J. Okuzono, R. Montazami and N. N. Hashemi, *Frontiers in Materials*, 2020, 7, 61.

2. J. Guo, A. E. Niaraki Asli, K. R. Williams, P. L. Lai, X. Wang, R. Montazami and N. N. Hashemi, Biosensors, 2019, 9, 112.

We appreciate the comment and we agree with the reviewer on the future steps of this work in terms of showing more data.

ii) The Raman spectra should be investigated in more detail. For example, the authors are encouraged to analyse D/G ratio and G band width which allows to draw (semi-quantitative) conclusions on edge and basal plane defectiveness (2D Mater. 2017, 4 (2), 025039). In addition the 2D band shape could be analysed to get a more quantitative picture on degree of exfoliation (at least as comparison); example procedures are described in various articles (Carbon 2015, 81 (0), 284-294, Nanoscale 2016, 8 (7), 4311-4323., Carbon 2013, 53, 357-365. Carbon 2020, 161, 181-189.). While all these methods have some restrictions, they would allow for a more solid sample comparison.

We have added the following analysis –

“The nanographite size (L_a) has usually been characterized using I_D/I_G . The qualitative control of structural transformation in various graphitic materials can be calibrated by Tuinstra and Koenig’s (T-K) empirical relation⁸. The amount of initial defects is given by L_a^{-1} and referred to as the defect density n_D . L_a is proportional to the quantity of disorder in nano crystallites. Furthermore, the defect distance in graphene containing 0-dimensional point defects can also be obtained through the correlations given below.

$$L_D^2 (\text{nm}^2) = \frac{(4.3 \pm 1.3) * 10^3}{E_L^4} \left(\frac{I_D}{I_G}\right)^{-1}$$
$$n_D (\text{cm}^{-2}) = (7.3 \pm 2.2) * 10^9 E_L^4 \left(\frac{I_D}{I_G}\right)$$

The above correlations can only be used under the assumption that graphene samples having point defects at distances satisfying $L_D \geq 10\text{nm}$ observed through excitation light within visible range^{9,10}.

*Based on the two equations given above and taking the laser energy to be 2.33 eV¹¹, our BSA-FLG samples with $\frac{I_D}{I_G} = 0.48$ contain L_D ranging between 19.89 nm and 14.56 nm. The n_D ranges between $7.21 * 10^{10} \text{cm}^{-2}$ and $1.34 * 10^{11} \text{cm}^{-2}$. We have not shown the L_D and n_D for the 90-hour milled samples since its $L_D < 10\text{nm}$, violating the above assumptions.”*

2. The plot of Raman D/G ratio as function of time in Figure 7a shows a significant amount of scatter. Is this related to spot to spot variations in each sample from the three measurements that were averaged? The authors are encouraged to average over more spots and include the standard deviation in the figure. For Figure 7b, they have chosen to show the data at 45h which follows a decreasing D/G ratio in the order no BSA, 1:10 BSA and 1:2 BSA. However, the plot above shows this is more or less a coincidence which varies for the different times. As such, it is not representative and it cannot be concluded that the D/G ratio is lower in the presence of BSA (as currently stated)

The samples were indeed averaged at different spots. Although we can't perform additional Raman characterizations, we agree with the reviewer that the decreasing D/G ratio seems like a coincidence and have removed it.

3. The sample preparation for each characterization should be included in the experimental section.

The experimental section has been modified with sample preparations included. It has been included below –

X-ray powder diffraction analysis – *“The powders of FLG were obtained by drying the suspension in an oven kept at 60 deg C. The as-dried powder was fixed on the zero-background silicon substrate using vacuum grease”*

Scanning Electron Microscopy – *“The powders of FLG were obtained by drying the suspension in an oven kept at 60 deg C. The as-dried powder was sprinkled sparingly on conductive carbon tape”*

Transmission Electron Microscopy – *“The sample preparation included pipetting 2 μ L of the aqueous FLG dispersion onto a carbon film copper grid. The excess material was removed using a filter paper and a thin film of the material was used for characterization.”*

Cell Imaging – *“The samples were prepared using BSA-FLG at varying concentrations. Propidium Iodide staining was conducted to identify cell death.”*

References –

- 1 J. I. Paredes and S. Villar-Rodil, *Nanoscale*, 2016, **8**, 15389–15413.
- 2 P. Laaksonen, M. Kainlauri, T. Laaksonen, A. Shchepetov, H. Jiang, J. Ahopelto and M. B. Linder, *Angew. Chemie*, 2010, **122**, 5066–5069.
- 3 V. Castagnola, W. Zhao, L. Boselli, M. C. Lo Giudice, F. Meder, E. Polo, K. R.

- Paton, C. Backes, J. N. Coleman and K. A. Dawson, *Nat. Commun.*, 2018, **9**, 1–9.
- 4 A. M. Gravagnuolo, E. Morales-Narváez, S. Longobardi, E. T. Da Silva, P. Giardina and A. Merkoçi, *Adv. Funct. Mater.*, 2015, **25**, 2771–2779.
- 5 K. S. Suslick and D. J. Flannigan, *Annu. Rev. Phys. Chem.*, 2008, **59**, 659–683.
- 6 W. B. McNamara, Y. T. Didenko and K. S. Suslick, *Nature*, 1999, **401**, 772–775.
- 7 E. B. Flint and K. S. Suslick, *Science (80-.)*, 1991, **253**, 1397–1399.
- 8 F. Tuinstra and J. Lo Koenig, *J. Chem. Phys.*, 1970, **53**, 1126–1130.
- 9 B. Li, Y. Nan, P. Zhang and X. Song, *Rsc Adv.*, 2016, **6**, 19797–19806.
- 10 L. G. Cançado, A. Jorio, E. H. M. Ferreira, F. Stavale, C. A. Achete, R. B. Capaz, M. V. de O. Moutinho, A. Lombardo, T. S. Kulmala and A. C. Ferrari, *Nano Lett.*, 2011, **11**, 3190–3196.
- 11 Photon Energies, http://www.dmp Photonics.com/Photon_energy/Photon_energies.htm.